# Melanogenesis and the Targeted Therapy of Melanoma

**DOI:** 10.3390/biom12121874

**Published:** 2022-12-14

**Authors:** Cang Li, Le Kuai, Rutao Cui, Xiao Miao

**Affiliations:** 1Skin Disease Research Institute, Zhejiang University School of Medicine, Hangzhou 310058, China; 2The 2nd Hospital and School of Medicine, Zhejiang University School of Medicine, Hangzhou 310003, China; 3Research Center for Life Science and Human Health, Binjiang Institute of Zhejiang University, Hangzhou 310053, China; 4Department of Dermatology, Yueyang Hospital of Integrated Traditional Chinese and Western Medicine, Shanghai University of Traditional Chinese Medicine, Shanghai 200437, China; 5Institute of Dermatology, Shanghai Academy of Traditional Chinese Medicine, Shanghai 201203, China; 6Innovation Research Institute of Traditional Chinese Medicine, Shanghai University of Traditional Chinese Medicine, Shanghai 200437, China

**Keywords:** melanoma, melanin, the targeted therapy of melanoma

## Abstract

Pigment production is a unique character of melanocytes. Numerous factors are linked with melanin production, including genetics, ultraviolet radiation (UVR) and inflammation. Understanding the mechanism of melanogenesis is crucial to identify new preventive and therapeutic strategies in the treatment of melanoma. Here, we reviewed the current available literatures on the mechanisms of melanogenesis, including the signaling pathways of UVR-induced pigment production, MC1R’s central determinant roles and MITF as a master transcriptional regulator in melanogenesis. Moreover, we further highlighted the role of targeting BRAF, NRAS and MC1R in melanoma prevention and treatment. The combination therapeutics of immunotherapy and targeted kinase inhibitors are becoming the newest therapeutic option in advanced melanoma.

## 1. Introduction

The pigmentation of hair, skin and eyes is predominantly dependent on the amount and composition of melanin [1]. Skin pigmentation is responsible for photoprotection [2]. Melanogenesis is the process by which melanocytes produce the pigment melanin in melanocytes [3]. Melanin is synthesized by tyrosine in melanocyte cells and then transported to keratinocytes via melanosomes and melanocytic dendrites [4]. Melanosomes are organelles that synthesize, store and transport melanin. Several internal and external factors, including genetics, ultraviolet radiation, endocrine, inflammatory and medications, function in melanogenesis. Skin pigmentary disorders such as vitiligo and melasma are correlated with the abnormal regulation of melanogenesis [3,5]. Melanin production is a unique character of melanocytes. Some signaling pathways in melanin production are also crucial in melanomagenesis. Thus, understanding the mechanism of melanogenesis is crucial to identify new preventive and therapeutic targets of melanoma.

## 2. Pigment Production

### 2.1. Melanocytes and Melanin

Melanocytes originate from the neural crest melanoblasts. Melanoblasts migrate to the basal layer of the epidermis and the hair follicles via a dorsolateral pathway after neural tube closure to develop melanocytes. This process is mainly regulated by the bone morphogenic protein (BMP) and Wnt signals [6]. When melanoblasts reach their final destinations and transform into melanocytes, they acquire the ability to produce melanosomes containing melanin [7]. Pheomelanin (reddish/brown) and eumelanin (brown/black) are two major types of melanin that differ in color and methods of synthesis. The ratio of eumelanin and pheomelanin determines, in part, the visible skin and hair color [8,9]. The size of melanosome is another part which determines our skin color. Regardless of the degree of pigmentation, the human epidermis contains approximately 74% eumelanin and 26% pheomelanin, and people with lighter skins have a low eumelanin level [9]. In general, eumelanin is photoprotective for pigmented tissues, whereas pheomelanin is phototoxic [10]. Tyrosinase (TYR), the key kinase for the synthesis of both eumelanin and pheomelanin, undergoes tyrosine hydroxylation to L-3,4-dihydroxyphenylalanine (DOPA) and then oxidizes rapidly to DOPAquinonel (DQ) [11]. In addition, tyrosine-related protein 1 (TRP1) can oxidize DHI-2-carboxylic acid (DHICA), which is derived from dopachrome and generated by tyrosine-related protein 2 (TRP2/DCT) to form eumelanin [12,13,14]. The most widespread form of albinism, oculocutaneous albinism type 1 (OCA1), is caused by TRY mutations [15]. Similarly, mutations and a lack of expression of *TRP1* are found in OCA3 [16]. Both the *TRP1* gene expression and TRP1 protein level are considerably downregulated in the lesional epidermis of vitiligo patients compared with non-lesional epidermis [17]. The level of TRP2 is also positively correlated with the melanin content in vivo. The transfected melanocytes expressing TRP2 show an increase in melanin content [18]. Altogether, TRY, along with TRP1 and TRP2, plays a crucial role in melanin synthesis by forming a multienzyme complex within melanocytes. They all control the quantity and quality of melanin production in melanocytes.

### 2.2. Melanosomes

Melanosomes are lysosome-related organelles and are found in most vertebrates and certain fungi. A melanosome is a site to synthesize, store and transport melanin [19]. The production of pigment melanin is a four-step process in melanosome [3]. In Stage I, vacuoles devoid of pigment are constructed inside a fibrillar matrix composed of glycoproteins. In Stage II, TYR, TRP1 and TYRP2 are received by melanosome with a structured, organized fibrillar matrix. Melanin synthesis and protein fibril deposition begin in Stage III. In the final stage (Stage IV), melanosomes develop into fully melanized cells [20]. Premelanosomal protein (PMEL17), a type I transmembrane glycoprotein, is a significant structural and biogenetic component of melanosome fibrillar structures in Stages I and II [21,22]. Mice with *pmel17* mutations have progressive coat color dilution [23]. Melanoma antigen recognized by T cells 1 (MART1), a melanoma-specific antigen and a melanosome-specific marker, forms a complex with PMEL17 to induce the expression, stability and trafficking of PMEL17. MART-1 siRNA inhibits PMEL17 processing in vitro to repress the PMEL17 expression [21]. In addition, different sizes of melanosome are found in human skin collected from different ethnicities. Studies have shown that melanosomes’ size is the largest in skin of African people and the smallest in the skin of European people [24].

Once melanosomes reach maturity in the epidermal melanocytes, they are transferred from the perinuclear region to the dendritic terminals [1]. They are then transported to keratinocytes, where the quantity and ratio of different melanin determines the skin color. Recent research indicated that keratinocytes produced from Caucasian skin exhibited higher autophagic activity than those derived from African-American skin, whereas melanosomes were more rapidly lost in keratinocytes derived from light skin as opposed to dark skin [25,26]. Keratinocytes, treated with lysosomal inhibitors or short interfering RNAs specific to autophagy-related proteins, accumulated more melanosomes [25]. Consistently, the melanin levels were greatly lowered by activators of autophagy and elevated by the inhibitors [25]. These results indicate that autophagy plays a pivotal role in regulating melanosome degradation in keratinocytes as well as melanocytes.

### 2.3. UVR-Extrinsic Activators and Regulators of Pigmentation

Numerous variables are known to regulate melanin formation, including ultraviolet radiation (UVR), cytokines and hormones. UVR is the most significant extrinsic factor in pigment production [3]. UVR comprises 5% of the solar radiation reaching the surface of the earth, and 95% UVA and 5% UVB, which are the main stimuli for photodamage, photoaging and skin cancer [27]. UV exposure and skin pigmentation are intimately connected each other. Most people living at lower latitudes (<388) acquire more eumelanin pigment in their skin as a result of significant UV exposure [28]. There are two types of tanning responses that depend on genetic backgrounds: immediate pigment darkening (IPD) and delayed tanning (DT) responses. IPD appears 5 to 10 min after exposure to UV, predominantly UVA, and disappears within minutes or days. Rather than an increase in melanin synthesis, the IPD response is mostly caused by the oxidation of pre-existing melanin and the translocation of melanosomes to the upper epidermal layers. The DT reaction occurs 3 to 4 days following UV exposure, mainly UVB, and diminishes within weeks as a result of an increased melanin production, especially eumelanin, which has a photoprotective effect [29].

There is growing evidence that even low doses of UVR could result in DNA photodamage and stimulate melanogenesis [30,31,32]. UVR, specifically UVB, induces DNA thymidine breaks and generates genotoxic cyclobutane pyrimidine dimers and 6-4 photoproducts in the skin. UVR also causes reactive oxygen species (ROS), which interact with multiple cellular components and lead to oxidative DNA damage [33]. On the one hand, UVR can directly induce the formation of diacylglycerol, a component of the melanocyte membrane that activates protein kinase C and, thus, regulates melanogenesis via tyrosinase phosphorylation [34,35]. On the other hand, UVR-induced DNA damage raises the levels of p53 in keratinocytes as well as in human melanocytes and melanoma cells [36,37] to transactivate the expression of pro-opiomelanocortin (POMC) preferentially in the epidermal keratinocytes [38]. The cleavage products of POMC, the α-melanocyte stimulating hormone (α-MSH) and adrenocorticotropic hormone (ACTH), are the agonists of melanocortin 1 receptors (MC1R), which transmit upstream signals to the cyclic adenosine monophosphate response element (CREB) and regulate the transcription of microphthalmia-associated transcription factor (MITF) through protein kinase A (PKA) signaling. MITF is the most important intrinsic factor and a key coordinator in many aspects of melanocyte [39] (Figure 1). Moreover, UVR exposure enhances the number of dendrites of the melanocytes and the transfer rate of melanosomes from the melanocytes to keratinocytes [40]. Eumelanin acts as a natural sunscreen against UVR-induced DNA damage [41]. There is an inverse association between the melanin content and the levels of cyclobutane pyrimidine dimers in the skin of persons with diverse skin pigmentations and ethnic origins, suggesting the photoprotective role of pigmentation [42]. As a result, people with fair skin are much more likely to get skin cancer [43].

### 2.4. MC1R, the Central Determinant of Pigmentation

MC1R is a member of the melanocortin (MC) receptor family, the smallest member of the G-protein coupled receptor (GPCR) class A (rhodopsin-like) family and is mainly expressed on melanocytes’ cell membranes [44,45]. MC1R is located on chromosome 16q24.3 and encodes a 7-pass transmembrane GPCRs of 317 amino acids [46]. MC1R signaling promotes melanin synthesis and melanosome transfer rate in melanocytes, making it one of the most important regulators involved in mammalian pigmentation [47]. Upon binding to melanocortins, including α-MSH and ACTH induced by UV exposure, MC1R stimulates adenylyl cyclase activity and then cyclic adenosine monophosphate-responsive (cAMP) production, activating a host of downstream signaling pathways such as PKA signaling [47,48] (Figure 1). The activation of PKA stimulates cAMP element-binding protein (CREB) phosphorylation and MITF activation, mediated by CREB [47]. In humans, MC1R is highly polymorphic. Approximate 420 variants of MC1R have been documented. Populations with lighter pigmentation showed more MC1R loss-of-function variants than those with darker pigmentation [49]. MC1R variants primarily reduced eumelanin synthesis, resulting in a “red hair color” (RHC) phenotype with red hair, fair skin and increased UVR sensitivity [50,51].

Furthermore, MC1R is a melanoma susceptibility gene, and the loss-of-function variants of MC1R have been found in 26–40% of melanoma patients [52,53]. The seven variants p.D84E, p.R142H, p.R151C, p.I155T, p.R160W, p. R163Q and p.D294H were revealed to be strongly linked with the development of melanoma [52]. It is plausible that as variants result in the loss of function of *MC1R*, this leads to a decline in eumelanin synthesis and, thus, leads to less effective protection against UV radiation and an increased risk of melanoma. In primary human melanocytes, the *MC1R* knockdown significantly impairs survival and DNA repair in response to UVR, hence increasing the risk of melanoma [54]. Therefore, controlling pigment formation by regulating MC1R may be an effective target for the prevention and treatment of melanoma.

The main antagonist for the MC1R competing with α-MSH is the agouti signaling protein (ASIP). Human *ASIP* expression in transgenic mice displayed yellow coat colors, and the expression of ASIP in human melanocytes inhibits tyrosinase activity and cell proliferation by blocking the stimulatory effects of α-MSH on cAMP accumulation and tyrosinase activity, respectively [55,56]. Notably, as Pomc1 knockout mice retain a dark coat color, it is possible that basal MC1R signaling is independent of ligands [57].

### 2.5. MITF, the Master Transcriptional Regulator of Pigmentation

Despite the identification of more than 100 loci involved in vertebrate pigmentation, the MITF is consistently a representative locus and a significant transcriptional regulator of pigmentation [58]. MITF is evolutionarily conserved and encodes for the basic helix–loop–helix leucine zipper (bHLH-Zip) transcription factor that belongs to the Myc-related family [59]. MITF belongs to the MiT transcription factor family alongside the transcription factors E3 (TFE3), EB (TFEB) and EC (TFEC), which are human oncogenes and have been implicated in melanoma [60]. The microphthalmia phenotype was first observed in mice with small eyes, white fur and deafness due to a homozygous mutant in *Mitf* [61], followed by subsequent abnormalities, including a decreased number of mast cells, faulty osteoclasts and an early onset of deafness due to multiple mutations in the *Mitf* locus [62]. The *Mitf* locus in humans is located on chromosome 3 and spans 229 kbp. Several major MITF isoforms, originating from distinct promoters, display different expression profiles, and the M-isoform of MITF (MITF-M) is a melanocyte-specific isoform involved in the regulation of pigmentation, melanocyte development and differentiation [39].

MITF-M is transcriptionally activated by CREB and numerous other transcription factors. ChIP-seq and analyses at single loci have identified that the transcription factors paired box gene 3 (PAX3), SRY (sex-determining region Y)-box 10 (SOX10) and lymphoid enhancer-binding factor 1 (LEF1) regulate the transcription of MITF directly in melanocytes [39,60]. Notably, the forced expression of MITF alone was insufficient to reprogram mouse and human fibroblasts. However, MITF, in conjunction with SOX10 and PAX3, is sufficient to reprogram the fibroblast into a functional melanocyte [59]. Post-translational changes in MITF, including phosphorylation, ubiquitination and SUMOylation, influence the activity and stability of the MITF protein [63].

Promoter–reporter analyses demonstrate that promoters of three primary pigmentation enzymes, TYR, TYRP1 and TYRP2, were transactivated by MITF directly. Additionally, MITF also induces the transcription of PMEL17 and MART-1 [64]. Other proteins involved in melanosome biogenesis and trafficking have also been identified as the direct MITF downstream targets. The G-protein coupled receptor 143 (GPR143) is a receptor for tyrosine, L-DOPA and dopamine that regulates melanosome maturation and size and has been identified as a direct target of MITF [65]. MITF has been reported to bind two E-boxes in the proximal region of the Rab27A promoter and stimulates its transcriptional activity, which ties MITF to melanosome transport for the first time [66]. In addition, MITF can also promote the expression of genes related to non-melanin synthesis, such as Tbx2 and BCL2 [67,68]. MITF is also involved in the cell cycle regulation of melanocytes. MITF leads to G1 cell cycle arrest by promoting the expression of the *p21* gene. The cooperation of RB1, the retinoblastoma protein, with MITF potentiates the ability of MITF to activate the transcription of *p21* [69].

### 2.6. Melanin Synthesis Process

Several melanin enzymes, including TYR, TYRP1 and TYRP2, are involved in melanin synthesis in melanocytes. First, L-tyrosine in melanocytes is oxidized to dopaquinone (DQ) by TYR. This step is a key speed-limiting step in the melanin formation process, and DQ is the substrate for the synthesis of eumelanin and pheomelanin [70]. Under normal physiological conditions, DQ can undergo self-cyclization to produce leucodopachrome, whereas leucodopachrome can further undergo redox reaction to produce dopachrome. After the decarboxylation of dopachrome, 5,6-dihydroxyindole (DHI) is obtained. Under the catalysis of TRP-2, 5,6-dihydroxyindole-2-carboxylic acid (DHICA) is further formed from dopachrome. Finally, DHI and DHICA undergo oxidative polymerization to form eumelanin [71] (Figure 2). In addition, during the synthesis of pheomelanin, DQ reacts with cysteine to produce cysteine dopa, and further oxidative polymerization takes place to produce soluble polymer pheomelanin [72].

## 3. The Targeted Therapeutics of Melanoma

Melanoma develops from human pigment cells, mainly occurs in the skin and is the most deadly and aggressive skin cancer [73]. In skin melanoma, based on the extent of cumulative sun damage (CSD), malignant melanomas are classified into high-CSD, low-CSD and low-to-no-CSD melanoma [74]. In the Asian population, melanoma mainly occurs in mucosal and acral sites. The metastasis of melanoma cells to vital organs leads to the death of melanoma patients. The occurrence of melanoma is related to ultraviolet (UV) radiation and the increase in the tumor mutation load (TMB) of melanocytes; people with pale skin, red hair and others are at highest risk of developing melanoma, compared to other pigmentation types [75,76]. The high mutation rate of melanoma is largely attributed to the mutagenic effect of ultraviolet light. According to the whole-genome sequences, the genes with significant mutations in skin melanoma include BRAF, CDKN2A, NRAS and TP53, et al. The genes with significant mutations in acral melanoma include BRAF, NRAS and NF1 et al. [77,78]. The genomic classification into four sub-types based on the pattern of the most prevalent significantly mutated genes is mutant BRAF, mutant RAS, mutant NF1 and triple-WT (wild-type) [78]. In the past 10 years, melanoma treatment has made significant progress, from targeting oncogenic BRAF and MEK to checkpoint blockade immunotherapy [79,80,81,82,83,84,85,86]. Unfortunately, the clinical benefit of melanoma treatment associated with current immunotherapeutic and molecularly targeted agents is limited by resistance and tumor recurrence [80,87,88].

### 3.1. BRAFV600E Mutation

As a proto-oncogene, the RAF gene was originally isolated from the murine sarcoma virus isolate 3611 and avian MH2 retrovirus [89,90]. BRAF, encoding a serine/threonine protein kinase, is a key regulatory protein in the RAS/RAF/MEK/ERK mitogen-activated protein kinase (MAPK) signaling pathway, which governs normal cell proliferation, differentiation and survival (Figure 3) [91]. BRAF mutation occurs at an early stage of melanoma and drives the melanocyte malignant transformation. Mutant BRAF is diagnosed at a high frequency (about 50–80%) clinically [78,91,92]. Among several types of BRAF mutation, the BRAF V600E mutation constitutes over 90% of the total, followed by other BRAF mutations, including V600K and V600D et al. [93,94]. The BRAF V600E mutation mimics phosphorylation by inserting a negatively charged glutamic acid residue adjacent to the phosphorylation site within the catalytic domain to induce the kinase activity of BRAF, with the subsequent phosphorylation and activation of MEK1 and MEK2 (Figure 3) [91,95]. The BRAF V600E mutation in melanoma activates the MAPK pathway to trigger melanocyte cell aberrant proliferation, inhibits the aberrant melanocyte cell apoptosis and eventually promotes melanoma progression [96]. Furthermore, about 10% of melanoma patients may show an intertumorally discordant BRAF status, and about 15% of BRAF-mutated melanomas may show intratumor BRAF heterogeneity [97]. In addition, BRAF mutations rarely occur concomitantly with KRAS or NRAS mutations [98]. At present, BRAF inhibitors have been widely used in the clinical treatment of melanoma.

#### 3.1.1. The BRAFV600E-Targeted Therapy in Melanoma

Sorafenib (BAY 43-9006), an oral multikinase inhibitor targeting tumors and tumor vasculature, was the first clinical drug used in clinical treatment with BRAF^V600E^ or metastatic melanoma patients. Unfortunately, the clinical trial indicated that sorafenib administration did not improve any of the end points over the placebo and cannot be recommended in the second-line setting for patients with advanced melanoma [99,100]. Subsequently, the BRAF inhibitors PLX4032 (vemurafenib) and GSK2118436 (dabrafenib) were synthesized and used in the clinical treatment of BRAF mutation melanoma [84,85,101,102,103] (Figure 4). In 2011, vemurafenib was approved for patients with advanced metastatic melanoma with the BRAF V600E mutation using FDA [103]. Later, the BRAF inhibitor dabrafenib was also approved for melanoma patients with the BRAF mutation [104,105]. Unfortunately, about 15% of patients showed no response to BRAF inhibition, and, among responders, about 50% developed acquired resistance after a median of 6–8 months [106].

#### 3.1.2. Combined Therapy of BRAF and MEK Inhibitors in Melanoma

Compared with the BRAF inhibitor alone, the combined treatment of BRAF and MEK inhibitors has shown a better effect for melanoma patients [107]. Clinical studies show trametinib, an MEK inhibitor, improved rates of progression-free and overall survival among patients who had metastatic melanoma with a BRAF mutation compared with chemotherapy [83]. In view of drug resistance and other problems arising from the BRAF inhibitor alone, the combination treatment of MEK and BRAF inhibitors was a better clinical achievement for melanoma patients with the BRAF V600E mutation. Clinical studies indicated that a combination of dabrafenib and trametinib, compared with dabrafenib alone, improved the 16% overall response rate in previously untreated patients who had metastatic melanoma with BRAF V600E or V600K mutations [101,108]. Subsequent studies further showed that combined dabrafenib and trametinib, compared with BRAF inhibition alone, led to long-term benefits, delayed the emergence of resistance and reduced toxic effects in patients who had melanoma with the BRAF mutation [109,110,111,112]. In addition, dabrafenib plus trametinib, compared with vemurafenib monotherapy, also significantly improved the overall survival in previously untreated patients with metastatic melanoma with BRAF V600E or V600K mutations and without increased overall toxicity [84].

Apart from trametinib, cobimetinib, another inhibitor of MEK, shows promising antitumor activity when combined with vemurafenib in patients with advanced BRAFV600-mutated melanoma [113,114] (Figure 4). In addition, encorafenib plus binimetinib treatment improved the median progression-free survival of patients up to 7.6 months compared with vemurafenib monotherapy. This study indicates that encorafenib plus binimetinib represents a new treatment option for patients with BRAF-mutant melanoma [115]. Although the combined use of BRAF and MEK inhibitors has shown a better effect in the clinical treatment of melanoma patients, some clinical adverse reactions are quite common in combination schemes, such as vomiting, nausea, fatigue, headache, arthralgia and so on [105]. Meanwhile, treatment with BRAF plus MEK inhibitors in BRAF mutation patients acquiring resistance remains a significant problem [75,106].

#### 3.1.3. The Combined Therapy of BRAF V600E Inhibitor and Immune Checkpoint Inhibitors in Melanoma

The tumor microenvironment (TME) plays a critical role in regulating tumor development, growth, invasion and metastasis [116,117]. The activity of tumor-infiltrating lymphocytes (TILs) of melanoma is involved in melanoma progression and prognosis [88]. Immune checkpoint inhibitors (ICIs) are effective in activating TILs to inhibit melanoma growth [86]. Ipilimumab, nivolumab and pembrolizumab targeting ICIs, including CTLA-4 or PD1, have been widely used in melanoma treatment clinically. All these antibody drugs are effective in activating TILs to rebuild the immune response to tumors in patients with advanced or metastatic melanoma [81,82,118,119,120]. However, 40–60% of patients with melanoma are not sensitive to ICI treatment [80,121]. A BRAF inhibitor in combination with ICIs has demonstrated significant clinical efficacy in BRAF mutation melanoma. Preclinical models suggest that combining BRAF and MEK inhibitors with PD-1 blockade therapy was of benefit to a subset of patients with BRAF(V600)-mutated metastatic melanoma [85]. The advanced melanoma patients who received dabrafenib and trametinib together with the pembrolizumab triplet therapy improved with 5.7 months progression-free survival compared with the doublet therapy of dabrafenib, trametinib and placebo but with a higher rate of grade 3/4 adverse events [102].

BRAF V600E-targeted therapy has radically changed the therapeutic landscape of melanoma, both for the advanced and adjuvant settings. However, drug resistance is still a key obstacle to the thorough treatment of melanoma. In the future, breakthroughs in the field of tumor microenvironments and heterogeneity will bring hope to overcome acquired drug resistance. In addition, the combination of immunotherapy and other targeted kinase inhibitors will also bring new and effective treatment plans for the treatment of BRAF-mutant melanoma.

### 3.2. NRAS and Melanoma

Ras protein, a small membrane-bound guanine nucleotide binding GTPase, acts as a molecular switch between the inactive and active states of GDP binding [122,123]. Active RAS GTPases participate in a variety of cellular processes, including proliferation, differentiation, apoptosis, cell–cell and cell–extracellular matrix interactions [124,125]. All mammalian cells express three closely related Ras proteins, termed H-Ras, K-Ras and N-Ras, that promote oncogenesis when they are mutationally activated at codon 12, 13 or 61 [123,126]. Genetic mutations in RAS isoforms are among the most prevalent oncogenic alterations detected in around 16–25% of all cancers [126,127,128]. KRAS or HRAS mutations are detected infrequently in approximately 5% of melanoma patients, whereas NRAS mutations (NRAS^mut)^ are found in about 25% of patients, which makes NRAS the second most frequent mutation type after BRAF in melanoma [129,130,131].

NRAS^mut^ melanoma is mainly caused by oncogenic missense mutations at codons 12, 13 or 61 [78]. Compared with oncogenic alterations at codons 12 or 13 which impair mechanisms of GTP hydrolysis, NRASQ61 occurs in 90% of all NRAS^mut^ melanomas and induces constitutive RAS-GTPase activity and conformational changes towards the GTP-bound active state [125,132]. NRAS^mut^ melanomas often display a dysregulated cell cycle, which is characterized by the upregulation of cyclin D1 and loss of tumor suppressor p16INK4A [130,133]. NRAS^mut^-mediated downstream effectors are the mitogen-activated protein kinase (MAPK) pathway and the phosphoinositide 3-kinase (PI3K)/protein kinase B (AKT) cascade (Figure 3) [122]. In comparison with BRAF^mut^, NRAS^mut^ melanomas are more aggressive with a lower median overall survival [134,135]. Patients with NRAS^mut^ melanomas have a poorer prognosis due to the high aggressiveness of RAS^mut^ tumors, lack of efficient targeted therapies and rapidly emerging resistance to existing treatments [128]. Understanding how NRAS-driven melanomas develop therapy resistance is crucial to develop new effective therapeutic strategies for patients with this kind of melanoma. At present, the treatment of NRAS mutant melanoma mainly focuses on the inhibition of the NRAS signal pathway-related kinase protein.

#### 3.2.1. The Targeted Therapy in Melanoma with NRASmut

Due to their picomolar affinity to bind GTP and the lack of druggable pockets outside of the nucleotide-binding site, it is hard to develop efficient GTP-competitive drugs that directly target RAS proteins [125,136]. To date, ICIs have been used in patients with NRASmut melanoma. However, the immunosuppressive microenvironment in NRASmut melanoma has been shown to limit the effects of ICIs [137,138]. Other treatment options include MEK inhibition in patients with NRAS^mut^ melanoma. Binimetinib represents a new treatment option for patients with NRAS-mutant melanoma after the failure of immunotherapy [128,139]. Oncolytic viral therapy (talimogene laherparepvec, T-VEC) is another new treatment option in patients with NRAS^mut^ melanoma. Furthermore, the combination of anti-PD-1 immunotherapies with oncolytic viruses yielded positive results [137,140].

#### 3.2.2. The Targeted Inhibition of NRAS Signaling in Melanoma

In addition toMEK inhibitors, inhibitors of other key proteins of the NRAS signaling pathway also repress the proliferation of melanoma with NRAS^mut^. ERK-mediated MAPK signal pathway reactivation often occurs after inhibiting BRAF and MEK in melanoma treatment [141]. Thus, the co-inhibition of MEK and ERK effectively reduced the growth of NRAS^mut^ melanoma [142,143]. ERK and MEK co-targeted inhibition is a useful treatment approach for patients with NRAS^mut^ melanoma. The activation of the MAPK pathway in NRAS^mut^ melanoma is achieved through the activation of the NRAS effector CRAF [129,144]. The disruption of CRAF-mediated MEK activation is required for effective MEK inhibition in KRAS^mut^ melanoma [145]. Altogether, the pan-RAF inhibitor demonstrated improved antitumor effects in patients [146]. Therefore, pan-RAF and MEK inhibitor combined therapy is bringing hope to the disease treatment of NRAS^mut^ melanoma. However, it is also reported that CRAF ablation does not affect tumor progression in NRAS^mut^ melanoma due to a rapid switch to BRAF-driven activation [147]. In addition, the inhibition of the PI3K-AKT signaling pathway is another promising option to prevent or delay the acquired resistance to the MAPK inhibitor in NRAS^mut^ melanoma [148,149].

The combination of several inhibitors, including MEK inhibitors, has been proved to be effective in NRAS^mut^ melanoma cells, such as the combination of MEK and CDK4/6 inhibitors [150]. The synergetic effects of the Rho/MRTF pathway inhibitor CCG-222740 and MEK inhibitor have also been demonstrated effectively in NRAS^mut^ melanoma cells [151]. The co-administration of the autophagy inhibitor chloroquine and MEK inhibitor has been tested in the treatment of RAS^mut^ cancers, including NRAS^mut^ melanoma [152,153]. The polo-like kinase 1 (PLK1), which is required for mitotic entry and centrosome maturation in late G2 phase/early prophase, is overexpressed in NRAS^mut^ melanoma. The MEK and PLK1 inhibitors synergistically induced melanoma cells apoptosis which is mediated by p53 signaling [154]. The MEK/PLK1 inhibitor combination might deserve to be further tested in patients with NRAS-driven melanoma. The combination of metabolic pathway-related kinase, such as phosphoglycerate dehydrogenase (PHGDH) and pyruvate dehydrogenase kinase (PDK), and MEK inhibitors also inhibited the growth of NRAS^mut^ melanoma cells [155,156]. However, most of these studies are only evaluated in vitro. Further studies are required to test these experimental combination treatment effects for NRAS^mut^ melanoma.

Serine/threonine kinase STK19 was identified as a novel NRAS activating protein. STK19 was found to have recurrent and potentially targeted mutations in large-scale melanoma exome data analysis [92]. STK19 phosphorylates NRAS to enhance its binding to its downstream effectors and promote the oncogenic NRAS-mediated malignant transformation of melanocytes [157] (Figure 5). STK19 targeted inhibitors was effective in blocking the oncogenic NRAS-driven malignant transformation of melanocytes and melanoma growth in vitro and in vivo [157]. This study provides a new and feasible therapeutic strategy for melanoma carrying NRAS mutations.

### 3.3. MC1R and Melanoma

MC1R encodes a cyclic adenylate-stimulated G protein coupled receptor that binds with its ligand, the alpha-melanocyte stimulating hormone (a-MSH), in pigment production after UV exposure [51]. MC1R is associated with skin sensitivity to sunlight and is recognized as a tumor suppressor of melanoma [158,159]. People carrying MC1R variants have a higher tendency to develop melanoma [159,160]. Its sequence variation can lead to partial or complete loss of receptor capacity [161]. The MC1R germline variation increases the risk of BRAF-mutant melanoma [162]. Thus, MC1R-targeted activation is an effective strategy in melanoma prevention, especially in individuals with red hair and fair skin [163].

#### MC1R Protein Palmitoylation Modification

Palmitoylation is often identified in GPCR protein, in which palmitic acid is reversibly added to the cysteine residues of the C-terminal tail or the intracellular loop. Protein palmitoylation modification profoundly affects the protein structure, stability, membrane localization and interaction with partner proteins [50,164]. Activating the palmitoylation modification of the MC1R protein has been demonstrated as a preventive and treatment strategy in redhead melanoma [50] (Figure 6). The pharmacological activation of palmitoylation in MC1R rescued the defect of the MC1R RHC variant and prevented melanomagenesis [50,163]. MC1R generates derived peptides during the expression of melanoma cells to induce a cytotoxic T lymphocyte (CTL) response [165,166]. These results suggest that MC1R-derived peptides show the potential to be developed into a melanoma vaccine for immunocellular therapy.

### 3.4. Other Therapeutical Target in Melanoma

In addition to the above therapeutic targets, other potential therapeutic targets are being found. Among them, the enzyme nicotinamide N-methyltransferase (NNMT) is a promising therapeutic target. Different studies demonstrated that NNMT is overexpressed in melanoma specimens, including cutaneous and oral melanoma [167,168,169]. In addition, the knockdown of NNMT expression in melanoma cell lines significantly reduces cell proliferation and migration, and the inhibition of NNMT enzyme activity increases the sensitivity of melanoma cells to dacarbazine treatment. NNMT is being recognized as a promising treatment target of melanoma [170]. Several NNMT inhibitors have been developed and tested in vitro [171,172].

## 4. Summary

Due to frequent exposure to the natural environment, skin has become the most common place where human cancer occurs. Melanoma caused by UVR is often considered one of the most aggressive and treatment-resistant human cancers. The pigment synthesis in melanocytes not only affects the normal skin coloring but also correlates with the development of melanoma. Understanding the molecular mechanism of pigment production will provide new preventive and therapeutic strategies for the treatment of melanoma.

In the past ten years, several breakthroughs have been developed in melanoma targeted therapeutics, including BRAF V600E-targeted inhibitors and ICIs. However, nearly half of patients are still not sensitive to these new therapeutic strategies. The acquired resistance is usually diagnosed in the sensitive patients. Thus, new therapy targets need to be explored and identified. Furthermore, different combination strategies should be further tested.

## Figures and Tables

**Figure 1 biomolecules-12-01874-f001:**
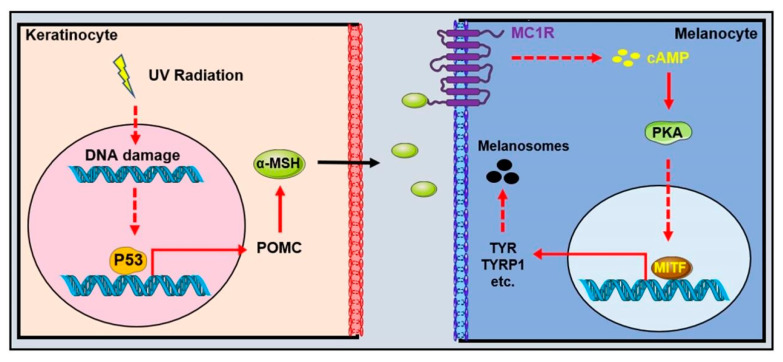
Schematic diagram of melanin production induced by ultraviolet radiation. UVR-induced P53 upregulation activates the transcription of opioid cortisol (POMC) in keratinocytes. POMC protein is then cleaved to form α-melanocyte stimulating hormone (α-MSH). α-MSH binds to its receptor, the melanocortin 1 receptor (MC1R), on cell membrane of melanocytes to induce the production of a circular AMP that activates the microphthalmic-associated transcription factor (MITF)-induced melanin enzymes’ transcription, including TYR, DCT and so on.

**Figure 2 biomolecules-12-01874-f002:**
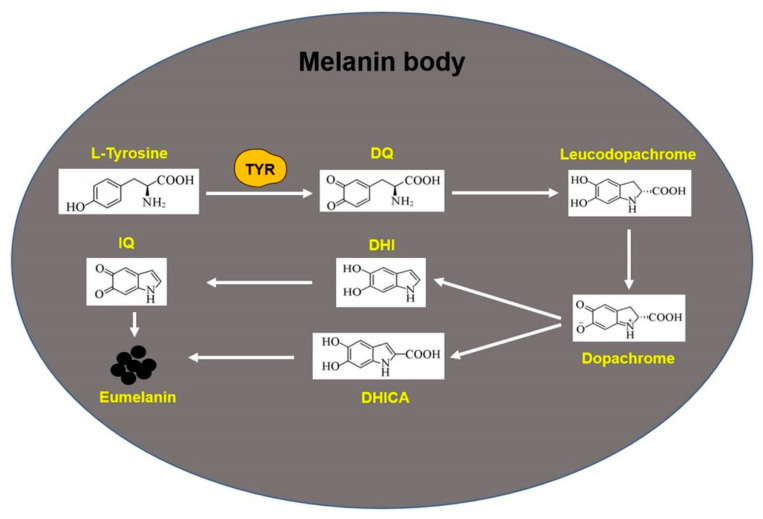
Biosynthesis process of eumelanin. TYR: tyrosinase; DQ: dopaquinone; DHI: 5,6-dihydroxyindole; DHICA: 5,6-dihydroxyindole-2-carboxylic acid; IQ: indole-5,6-quinone.

**Figure 3 biomolecules-12-01874-f003:**
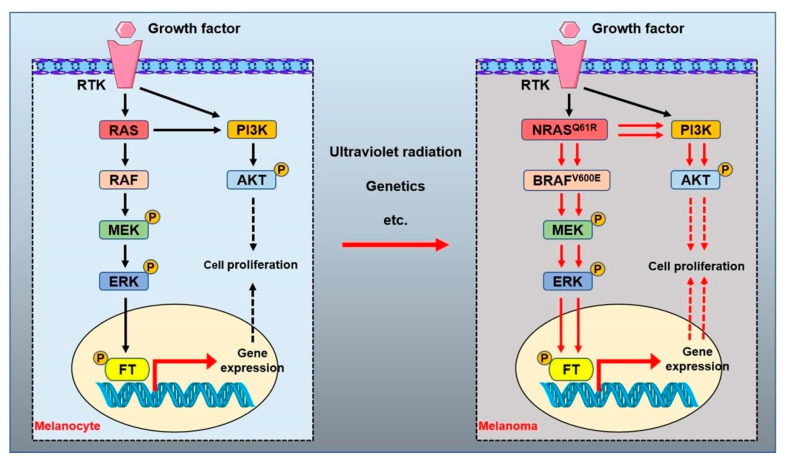
RAS signal pathways in melanoma. The mitogen-activated protein kinase (MAPK) and PI3K–AKT signaling pathways regulated by RAS permit the balanced control of basic cellular functions in melanocytes. Numerous factors are related to the transformation of normal melanocytes into melanoma, including genetics and ultraviolet radiation. The mutant NRAS or BRAF activates the MAPK and PI3K–AKT signal pathways to promote the occurrence and proliferation of malignant melanocytes in melanoma. RTK: receptor tyrosine kinase. ERK: extracellular signal-regulated kinase. MEK: mitogen-activated protein kinase. AKT: protein kinase B. PI3K: phosphoinositide-3 kinase. FT: transcription factor. P: phosphorylated.

**Figure 4 biomolecules-12-01874-f004:**
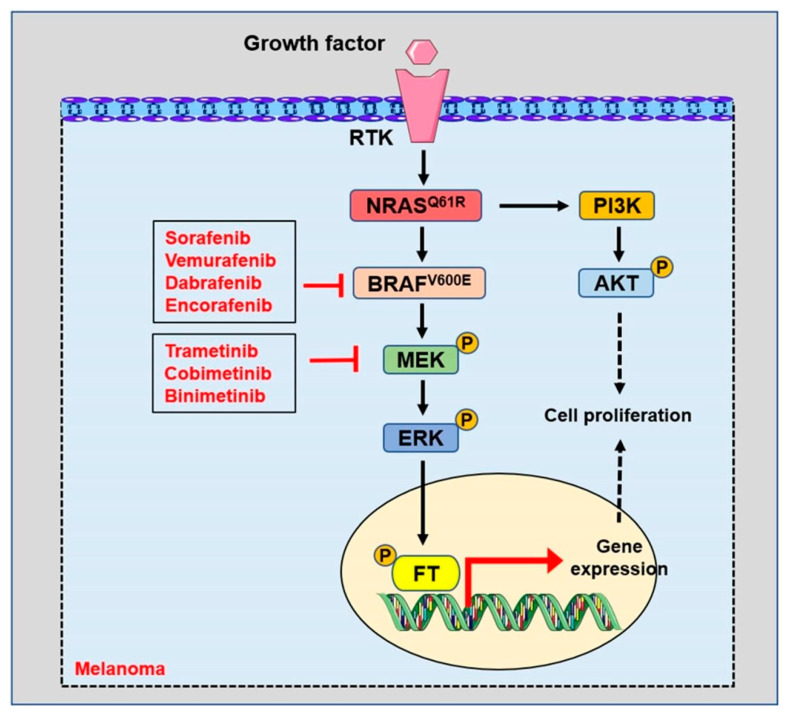
BRAF and MEK inhibitors are used to treat patients with melanoma. ERK: extracellular signal-regulated kinase. MEK: mitogen-activated protein kinase. AKT: protein kinase B. PI3K: phosphoinositide-3 kinase. FT: transcription factor. P: phosphorylated.

**Figure 5 biomolecules-12-01874-f005:**
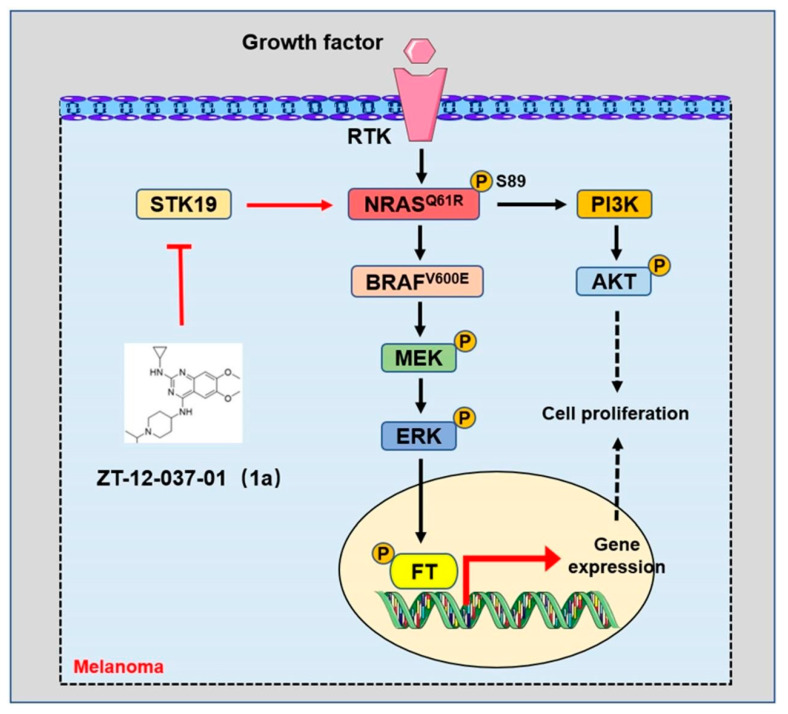
Targeted inhibition of STK19 inhibits oncogenic NRAS-driven melanomagenesis. In melanoma, STK19 phosphorylates NRAS at the serine 89 site to increase the activation of NRAS-regulated signaling pathways. ZT-12-037-01 (1a) is a specific STK19-targeted inhibitor, which blocks the malignant transformation of melanocytes and melanoma growth driven by carcinogenic NRAS by inhibiting STK19.

**Figure 6 biomolecules-12-01874-f006:**
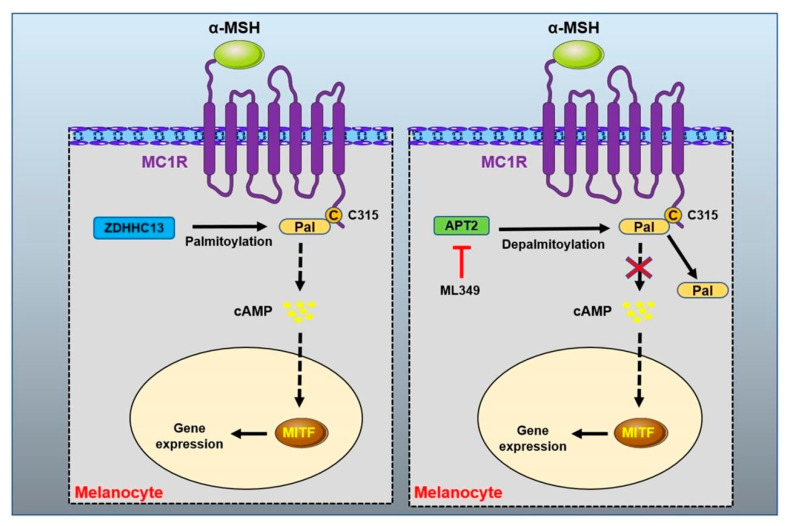
Palmitoyl modification of MC1R. The protein acyltransferase ZDHHC13 mediates the palmitoylation modification at cysteine 315 site of MC1R protein to activate MC1R signal. APT2 identified as MC1R depalmitoylase, and the treatment of APT2 inhibitor ML349 can effectively increase MC1R signal transduction and inhibit UVB-induced melanoma.

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
