# Peer review of "Melanogenesis and the Targeted Therapy of Melanoma"

_biomolecules, 2022, doi:10.3390/biom12121874_

Round 1

Reviewer 1 Report

 The review article titled "Melanogenesis and targeted intervention of melanoma" by Li et al., is an interesting article. The authors have done a nice job describing transcriptional regulation of melanin production, etc. However, there is a sudden shift in the direction of the paper and the authors focus on various signaling molecules and immunotherapies. In order to prevent reader's confusion and stay on track, I recommend that the authors emphasise on one of the three themes developed in the current manuscript (melanin, signaling molecules, therapies)  

Author Response

Reviewer #1 Comments for the Author:

The review article titled "Melanogenesis and targeted intervention of melanoma" by Li et al., is an interesting article. The authors have done a nice job describing transcriptional regulation of melanin production, etc. However, there is a sudden shift in the direction of the paper and the authors focus on various signaling molecules and immunotherapies. In order to prevent reader's confusion and stay on track, I recommend that the authors emphasise on one of the three themes developed in the current manuscript (melanin, signaling molecules, therapies

Response: Thanks for the positive comments and constructive suggestion. In this review manuscript we main reviewed the mechanisms of melanogenesis and highlighted the targeted intervention of melanoma. melanin production pathway plays a crucial role in the occurrence of melanoma. Following the reviewer’s suggestion, we further emphasized that understanding the mechanism of melanogenesis is crucial to identify new preventive and therapeutic target of melanoma. (please see revised manuscript).

Reviewer 2 Report

The manuscript “Melanogenesis and targeted intervention of melanoma” is a review article regarding the current available studies on the mechanisms of melanogenesis and newest therapeutic option for melanoma.

The manuscript has several important concerns that must be addressed in order to consider the manuscript suitable for publication:

1.       In the abstract the sentence “Numerous factors are linked with the melanin production, including genetics, ultraviolet radiation (UVR) and inflammation, et al.” should be rephrased. Using “et al.” in this context is weird.

2.       In the text there is an excessive use of short sentences barely connected. The manuscript requires a language revision to improve the language and the construction of sentences.

3.       Line 26-36: it is not clear what this part is (an Introduction?). There is no title for this paragraph, neither a number to identify it.

4.       The quality of Figure 1 is poor and must be improved.

5.       The figure 2 seems to be prepared carelessly. The geometry of molecules is not respected due to stretched figures (e.g. IQ and dopachrome). Please revise the figure accordingly.

6.       Regarding the newest perspectives of therapeutic options, authors completely ignored the enzyme nicotinamide N-methyltransferase (NNMT), a promising therapeutical target in melanoma. Different studies demonstrated that NNMT is overexpressed in melanoma specimens, both cutaneous and oral melanoma. Furthermore, its downregulation in melanoma cell lines triggered a decrease in cell proliferation and was associated with an increased sensitivity to treatment with dacarbazine, thus representing a promising therapeutical target.

A paragraph summarizing all these studies is required. This is particularly important since several NNMT inhibitors have been developed, which could be tested on melanoma cells (PMID: 34704059 and PMID: 36288465).

Minor:

Line 349: two dots are present.

Author Response

Reviewer #2 Comments for the Author:

The manuscript “Melanogenesis and targeted intervention of melanoma” is a review article regarding the current available studies on the mechanisms of melanogenesis and newest therapeutic option for melanoma.

The manuscript has several important concerns that must be addressed in order to consider the manuscript suitable for publication:

Response: Thanks for your comments.

  1. In the abstract the sentence “Numerous factors are linked with the melanin production, including genetics, ultraviolet radiation (UVR) and inflammation, et al.” should be rephrased. Using “et al.” in this context is weird.

Response: Thanks for the careful review. We apologize for our mistakes in writing and we change it in the manuscript.

  1. In the text there is an excessive use of short sentences barely connected. The manuscript requires a language revision to improve the language and the construction of sentences.

Response: Thanks for your valuable suggestions. According to your modification comments, we will revise the manuscript language by using the language modification service. At present, we are consulting relevant institutions and will complete the language modification as soon as possible.

  1. Line 26-36: it is not clear what this part is (an Introduction?). There is no title for this paragraph, neither a number to identify it.

Response: Thanks for the careful review. This part is introduction and we have added the “Introduction” title before the paragraph (Please see Line 34 in the revised manuscript).

  1. The quality of Figure 1 is poor and must be improved.

Response: Thanks for the suggestions, we change Figure 1 in the revised manuscript.

  1. The figure 2 seems to be prepared carelessly. The geometry of molecules is not respected due to stretched figures (e.g. IQ and dopachrome). Please revise the figure accordingly.

Response: Thanks for your careful review. We are apologized for our negligence. we have replaced Figure 2 in the revised manuscript.

  1. Regarding the newest perspectives of therapeutic options, authors completely ignored the enzyme nicotinamide N-methyltransferase (NNMT), a promising therapeutical target in melanoma. Different studies demonstrated that NNMT is overexpressed in melanoma specimens, both cutaneous and oral melanoma. Furthermore, its downregulation in melanoma cell lines triggered a decrease in cell proliferation and was associated with an increased sensitivity to treatment with dacarbazine, thus representing a promising therapeutical target.

A paragraph summarizing all these studies is required. This is particularly important since several NNMT inhibitors have been developed, which could be tested on melanoma cells (PMID: 34704059 and PMID: 36288465).

Response: Thanks for your comments and constructive suggestion. we have added a paragraph to summarizing related research progress of NNMT in the manuscript (Line 455-463 in the revised manuscript).

Round 2

Reviewer 2 Report

The manuscript is improved and can be published.